# Developments in 3D Visualisation of the Rail Tunnel Subsurface for Inspection and Monitoring

**Thomas McDonald ***, **Mark Robinson** and **Gui Yun Tian**

Newcastle University Centre of Excellence for Mobility and Transport, Newcastle upon Tyne NE1 7RU, UK
* Correspondence: t.mcdonald@newcastle.ac.uk

**Featured Application: The review presented in this work has practical application to the conception, development and refinement of new technologies and visualisation frameworks pertaining to railway tunnel subsurface inspection. Subsequent application to the development of prototype self-sustaining digital twin tunnels also presents opportunity. In both cases, practical end user benefit would be improvement to the clarity and comprehensiveness of subsurface inspection datasets, better informing targeted maintenance strategy planning.**

**Abstract:** Railway Tunnel SubSurface Inspection (RTSSI) is essential for targeted structural maintenance. 'Effective' detection, localisation and characterisation of fully concealed features (i.e., assets, defects) is the primary challenge faced by RTSSI engineers, particularly in historic masonry tunnels. Clear conveyance and communication of gathered information to end-users poses the less frequently considered secondary challenge. The purpose of this review is to establish the current state of the art in RTSSI data acquisition and information conveyance schemes, in turn formalising exactly what constitutes an 'effective' RTSSI visualisation framework. From this knowledge gaps, trends in leading RTSSI research and opportunities for future development are explored. Literary analysis of over 300 resources (identified using the 360-degree search method) informs data acquisition system operation principles, common strengths and limitations, alongside leading studies and commercial tools. Similar rigor is adopted to appraise leading information conveyance schemes. This provides a comprehensive whilst critical review of present research and future development opportunities within the field. This review highlights common shortcomings shared by multiple methods for RTSSI, which are used to formulate robust criteria for a contextually 'effective' visualisation framework. Although no current process is deemed fully effective; a feasible hybridised framework capable of meeting all stipulated criteria is proposed based on identified future research avenues. Scope for novel analysis of helical point cloud subsurface datasets obtained by a new rotating ground penetrating radar antenna is of notable interest.

**Keywords:** railways; tunnel; subsurface; inspection; visualisation; ground penetrating radar; 360GPR; structural health monitoring; building information modelling; extended reality

## 1. Introduction

Railway tunnels provide critical transport links for passengers and freight through terrain otherwise impassable to trains, facilitating time-efficient navigation through mountains, under waterbodies and bypassing human-made obstructions (e.g., buildings, utilities, mass-transit routes). As confined high-traffic subterranean infrastructure, tunnels are inherently hostile and dangerous environments, suffering perpetual degradation from both environmental and human factors (e.g., shifting landmass, extreme weather, aboveground construction) [1–5] which seed discernible damage to the intrados—the innermost surface of the tunnel arch [6]—surface and subsurface. In the UK, unlike comparatively modern highway and metro tunnels, railway tunnels frequently date back to the Victorian era. These historic masonry structures are inherently weaker than their modern concrete

counterparts, meaning complex degradation can rapidly develop in the vicinity of seeded damage. Therefore, detection and hazard-level evaluation of all structural features (assets, defects) during Railway Tunnel Inspection (RTI) surveys is essential to inform targeted maintenance, ensuring continued safe and efficient operation. Use of Non-Destructive Inspection/Evaluation (NDI/E) techniques for Rail Tunnel SubSurface Inspection (RTSSI) is of paramount importance in modern surveys; however, they are not infallible due to accuracy and clarity limitations. Consequently, undetected seeding and growth of concealed subsurface defects can complicate or even scrub maintenance attempts, irrespectively posing serious safety risks that can endanger life. Timely reminders include undetected microfracture growth which caused catastrophic failure of the Gerrards Cross Tunnel (UK, 2005) [7,8] and two violent crown failures which partially collapsed 18 m of the Yangshang Tunnel (China, 2017) [9].

Collectively these dangers highlight urgent need for a comprehensive, reliable, repeatable, time-efficient and clear RTSSI visualisation framework, based on NDI techniques, for accurate intrados subsurface feature detection and evaluation. In this work, we review the effectiveness of current RTSSI-relatable visualisation frameworks, focusing on the increasing capabilities of realistic 3D surveys and discussion of future research opportunities. Our aims are to highlight common critical limitations of current strategies and propose viable, pertinent improvements. Following an overview of research methodology (Section 2) we explore leading NDI methods in RTI for RTSSI (Section 3), before deliberating the issues of applying heuristic comparisons (Section 4). From this, we formulate criteria for effective RTSSI visualisation frameworks (Section 5), then appraise the scope of current connected research efforts (Section 6). A discussion of trends and identified findings is finally presented (Section 7).

## 2. Materials and Methods

We analyse journal references, practical studies and commercial systems pertaining to RTSSI-relatable visualisation frameworks. For this purpose, we partition notion of an RTSSI visualisation framework into two sequential phases: (1) Data Acquisition Approaches; (2) Information Conveyance Schemes. A subtle remark, note that information is not itself raw data but the meaning from raw data. 'The tree holds 5 apples' is raw data, but knowing we expect it to hold 20 gives meaning to the data (i.e., the harvest is poor). Context and analysis turn raw data into useful information.

No prior review work encountered considers both described phases in detail. In fact, we found only two literary reviews directly related to RTSSI [10,11]. Both principally analyse literature concerning phase (1), of which [10] is the only dedicated review article. However, being published in 2015, it now lacks current relevance due to technological advancements. Discussions of ROBO-SPECT (RS) are the only RTSSI-specific scheme we do not consider outdated, but consideration will only be paid its most recent publications. The 2015 review provides a comparative 'baseline' for discerning small updates on already established methods from genuinely novel RTSSI innovations. Our review focuses on the latter as this provides greater benefit to current researchers and practitioners, with [10] providing historic reference. By contrast, we note although [11] is the more recent publication, it relies on many references previously provided in [10]. Our observations clearly necessitate creation of a novel review addressing both phases of RTSSI visualisation framework development, spanning research projects and commercial systems documented between 2015 and 2021.

Our review procedure is illustrated in Figure 1 and draws upon knowledge from over 300 information resources. We formally cite 255 primary resources (e.g., reviews, articles, proceedings, etc.). Upwards of 60 secondary resources (e.g., internal reports, cooperate multimedia, web-resources, etc.) provide further insight, however are not formally referenceable Owing to the fundamental dependency phase (2) has on phase (1), materials considered have frequently contributed to discussion of both areas. As a conservative estimate, the distribution of total literature is approximately 67% across phase (1) and

55% across phase (2), with 33% providing contextualisation. Maximal crossover is approximately 69%. Academic publications are obtained by 360-degree searches of journal articles and conference proceedings through databases including IEEE Xplore, Springer, MDPI and Google Scholar. Materials concerning commercial technology solutions are primarily sourced from the relevant corporate organisations' website, system manuals and any associated technical papers. Keywords recurringly searched include: "Railway Tunnels", "Structural Health Monitoring", "Subsurface", "Defects", "Computer Vision", "Visualisation", "Human Computer Interaction", "LiDAR", "Photogrammetry", "Ground Penetrating Radar" and "Extended Reality". For a detailed breakdown of key references, see Appendix A.

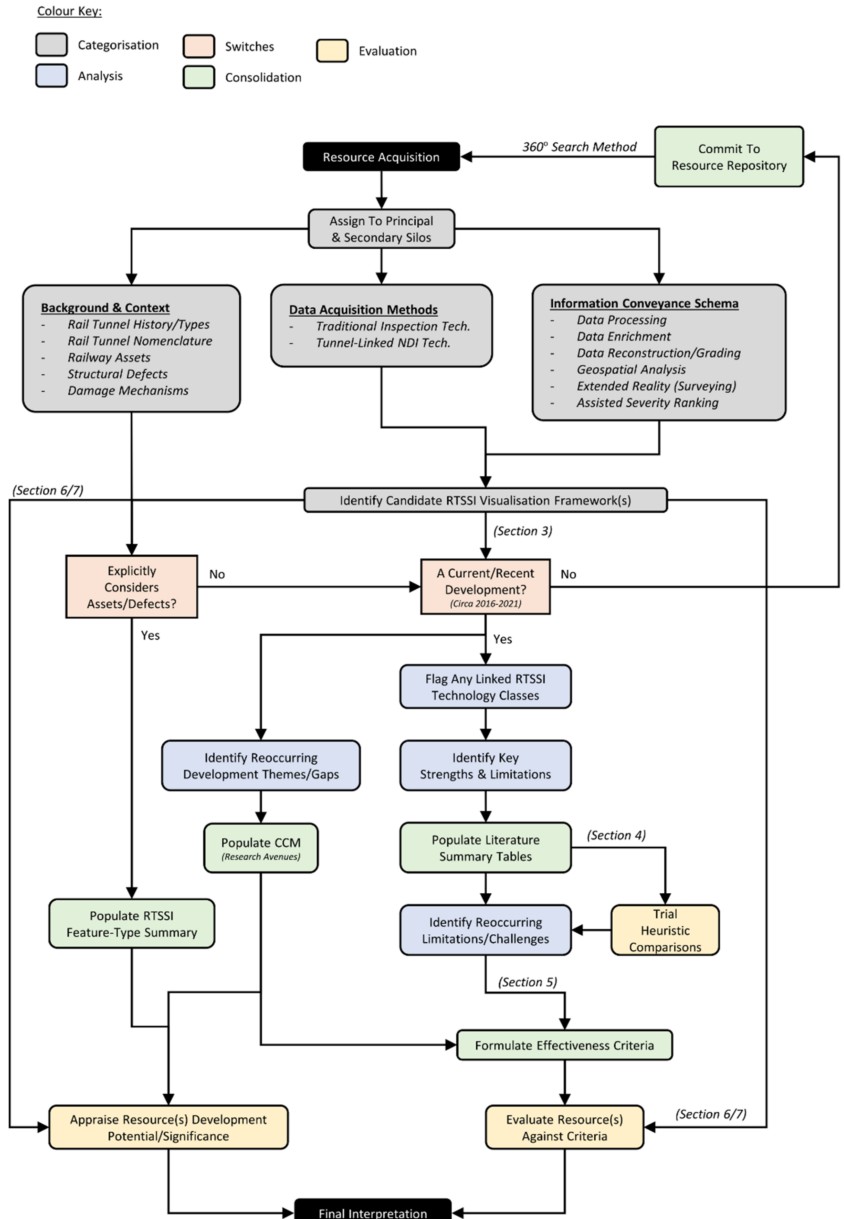

**Figure 1.** Flowchart of reviewing procedure prescribes handling, analysis and management of resources, alongside providing a rigorous procedure for appraising the effectiveness and scope of frameworks considered.

## 3. RTI Data Acquisition Methods

To gauge current state of the art in RTSSI-relatable visualisation frameworks, we first consider current leading subsurface data acquisition methods. Note that data acquisition must logically precede information conveyance in all conceivable frameworks by chronological reasoning.

### 3.1. Visual Methods

Visual assessment is the longest-established NDI method for RTI and is still widely adopted today, particularly across the UK and Chinese rail networks [12,13]. We subdivide methods into two classes: (i) Traditional and (ii) Modernised. Traditional evaluation is exclusively based on engineers' learnt association between visual indicators (e.g., workmanship inconsistencies, material fatigue hallmarks) and fault likelihood. Problematically, engineers infrequently share similar extents of practical experience, resulting in high subjectivity. Crosschecks and multi-pass surveys can partially reduce accuracy and consistency variations but take significantly longer to implement at increased resource cost, closure times and rail worker risk. Handwritten notetaking ambiguity, incompleteness and inherent susceptibility to human error also present issues for later analysis. They entice misinterpretation, causing unnecessary delays and disruption. However, being low expense and reasonably accurate (if performed by more experienced engineers), coupled with human-aptitude at informed predications from non-structural information (e.g., history of construction practices); traditional methods can time-efficiently localise visibly degraded quadrants requiring repair.

Modernised methods mostly utilise Close-Range Photogrammetry (CRP) to provide referenceable intrados imagery. Units commonly employ RGB optical cameras mounted on moving platforms for stability and time-efficiency. These include: Pushcarts/Rail-Trolley (RT), Road-Rail Vehicle (RRV) and Robotic Traction Unit (RTU). Merging resultant overlapping orthophotos via mosaicing [14,15] allows tunnels to be 'unwrapped'—permitting analysis in 2D—although we more frequently find studies adopt 3D CRP topography model reconstruction via 'Structure from Motion' (SfM) algorithms [16–19].

A noteworthy recent innovation includes 'Digital Imaging for Condition Asset Monitoring System' (DIFCAM) [20]; an RRV-mounted optical array deigned to reduce crew sizes and inspection durations. Although 2014 marks DIFCAM's last major study [21], scope of its successor project DIFCAM Evolution [22] discusses subsurface imaging and automated defect recognition technology integration. However, due Visual assessment is the longest-established NDI method for RTI and is still widely a lack of available details or recent publication activity, we reside this to speculation only. Of comparable interest, [23] presents a 'Moving Tunnel Profile Measurement' system (MTPM-1) which deploys a novel rotating camera for CRP that tracks a translating laser target to achieve swift 3D capture of a 100 m tunnel in 3 min. Use of a more lightweight camera is necessary for smoother rotation and reduction of prevalent lens-distortion.

Overall, visual methods provide extensive surface inspection prospects but are impractical for subsurface inspection since tunnel intrados' are opaque, except where defects have already exposed the subsurface. For a summary of defect types see Section 6.1.2 and consult 'Ring Separation and Debonding'. We believe proposed revisions of MTPM-1 show promise and would further benefit from fusion with automatous RTU locomotion described in [24] to facilitate 24/7 remote deployment.

### 3.2. Acoustic Methods

Subsurface features modify the characteristics of propagating soundwaves. Acoustic methods pulse predefined waveforms into the tunnel intrados and analyse resultant distortion and delay to identify audible indicators of defects. Acoustic methods can be subdivided into Ultrasonic Testing (UST) and Infrasonic Testing (IST). In UST, reductions in travelling pulse velocity correspond to elastic deformation of defected regions [25,26];

contrastingly for IST, defects are indicated by high resonant frequency components in returning pulses [27].

We only encountered two research groups directly applying UST to tunnel subsurface inspection. In [28], UST is extremely time-inefficient, requiring 9–25 min to scan 1m of tunnel wall and necessitating use of a preliminary GPR scan (Section 3.6) to localise suspected features. Likewise, despite robotic automation, UST scans performed by tunnel profiler ROBOSPECT achieve comparably inefficient durations of one hour to scan 6m [29] and are optimised for surface level crack and spall detection only [30,31]. Evidently, UST can be considered ill-suited for RTSSI, where surveys must be swift to minimise periods of tunnel closure.

IST proves more useful for RTSSI. Traditionally, hammer-strike emissions are performed by experienced human operatives who detect audible defect indicators 'by ear' alone, but those remaining are few, approaching retirement and are not being replaced. Faster robotic schemes are now preferential, boasting improved high level access achieved by mounting hammers to robotic arms [32–35] on Variable Guide Frames [36] and UAVs [37,38]. We notably uncovered a unique non-contact infrasonic UAV system [39] successfully inducing hammer-strike reminiscent flexural vibrations in infrastructure at distances of up to 5 m, for which application to remote-RTSSI presents an interesting research venture.

Detrimentally, inherent reliance on human interpretation of audio-spectra (which do not physically resemble subsurface features they convey) critically limits the insight non-specialist end-users can draw from IST without costly training or additional contextual metadata (e.g., maps of striking locations).

*3.3. Laser Methods*

Terrestrial Laser Scanning (TLS), also termed LiDAR (Light Detection And Ranging), utilises directed lasers to scan the visible tunnel intrados, generating dense 3D point clouds (Figure 2a) at up to $1 \times 10^6$ datapoints per second [40–43]. Visible light impulses reflect with variable intensity informing relative distances. However, datapoints lack classification labels and do not penetrate the subsurface. This makes segmentation of tunnel features challenging [44], but does permit direct insight into subsurface condition (e.g., profile distortions indicate abnormal strains) [45]. Pursuit of TLS integration with counterpart penetrating NDI methods marks an emerging avenue of long-term research. We note that the development of a standardised, efficient and reliable method to perform the essential alignment of multiple point cloud datasets—to form a unified digital environments—will be a key milestone for innovators to achieve before practical deployment becomes mainstream RTSSI practice.

Returning to standalone laser methods, we found TLS-RTI studies and commercial contractors most commonly deploy FARO® FOCUS scanning modules [46–49] (Figure 2b) or the Z+F Profiler® 9012 [50–52] to assimilate RGB optical photography for improved end-user navigational ease in recovered point clouds. Noteworthy innovations include an automated deformation detection assembly [11], which utilises a novel Circular Laser Scanning System (CLSS), highlighting the practicality of adopting circular sensing arrays that complement natural tunnel curvature.

Notable innovation is showcased in the Tunnel Monitoring and Measurement System (TMMS) developed by [52]. The prototype visualisation framework utilises a Z+F Profiler® 9012 mounted on a bespoke rail trolley (Figure 2c) to pass RGB LiDAR tunnel point clouds and a 'roaming video' feed of intrados condition to an engineer's tablet PC. The developed hardware bares strong similarities with a similar mobile TLS apparatus used in [49] employing a FARO X330 scanner. Validation trials in China's Zhengzhou Metro network demonstrate practical deployment capability but also relay that primary functions of ingress and cross-sectional deformation detection suffer noteworthy accuracy and stability reduction when applied to non-circular tunnel profiles (e.g., horseshoe, elliptical, etc.). TMMS therefore flags the importance adaptability in the design of new RTSSI solutions for

wide-scale deployment, particularly on older rail networks (e.g., UK) which adopt multiple 'standard' tunnel cross-section variants.

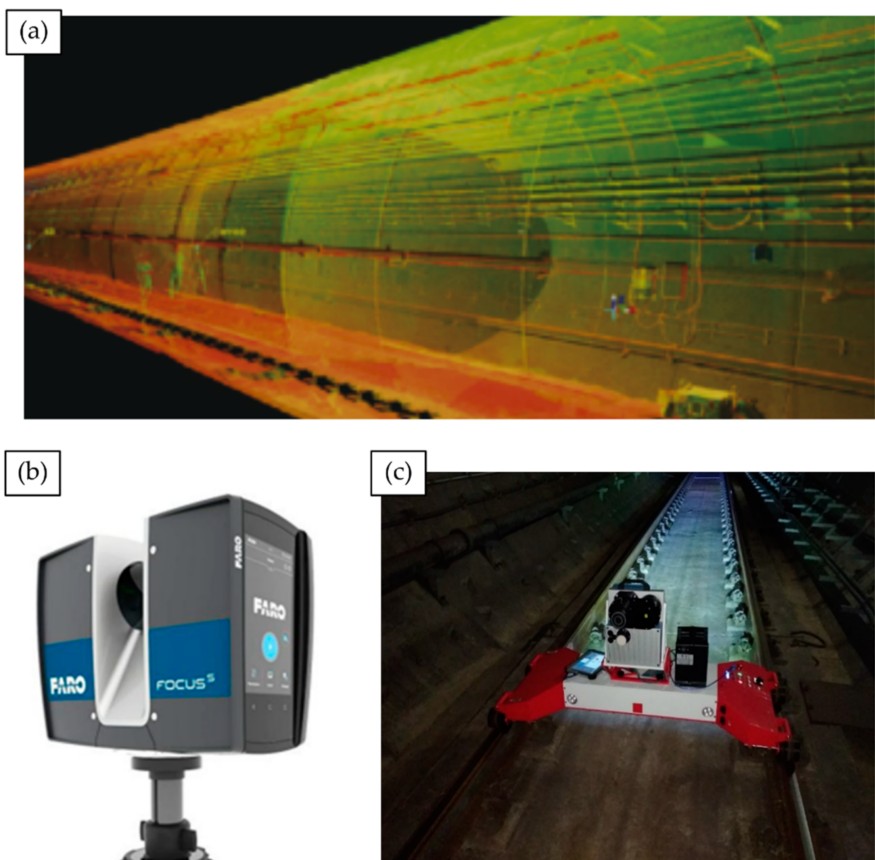

**Figure 2.** TLS for tunnel inspection. (**a**) 3D point cloud returned from a TLS metro tunnel survey [52]; (**b**) A FARO® FOCUS 350 scanning module, commonly deployed for infrastructure surveys; (**c**) TMMS rail-trolley transports a Z+F9012 to capture a TLS point cloud of a Zhengzhou metro tunnel.

### 3.4. Thermographic Methods

Subsurface faults modify thermal emission patterns of nearby interior tunnel surfaces, causing abnormal variations. Visualising temperature distribution profiles (Thermometry) facilitates localisation of suspected near-surface features (Thermography) [53,54], but recovery of specific attributes defers to higher quality UST or localised GPR imaging. Active Thermography (ACT) heats surfaces using halogen lamps [55], air guns [56] or inductive-heating elements [57] to induce exaggerated thermal responses. Abandoned testing by [58] and remarks of [59] affirm that heating element operation for RTSSI would incur impractical cost and could debond masonry, explaining its literary absence. We find use of infrared camera arrays for passive Infrared Thermography (IRT) more commonplace, owing to swifter and less costly implementation. Leading systems identify both air and water filled voids, with individual scans displayable as 2D panoramic imagery [53] or pioneering 3D mesh overlays on digital structural models rendered using TOSCA-FI [60] (Figure 3) or Augmented Reality [61] (Section 6.2). Despite recent work, Thermography still exhibits persistent limitations [54] undermining direct application to RTSSI:

- Results are highly sensitive to ambient temperature conditions which diminishes anomaly contrast (e.g., daily and seasonal variation);
- Thermal insulation and heat-resistant coatings used for tunnel temperate regulation and fire resilience can skew results. High thermal dissipation can easily restrict penetration d < 30 mm [62];

- Subsurface water content variation (e.g., increased permeation following rainfall or snow) can mask or exaggerate thermal profiles of faults;
- Enclosed, curved tunnel geometry restricts available viewing angles and confine results to 2D, even in a 3D mesh overlay, making inference of feature depth and physical form very challenging even for experienced operatives.

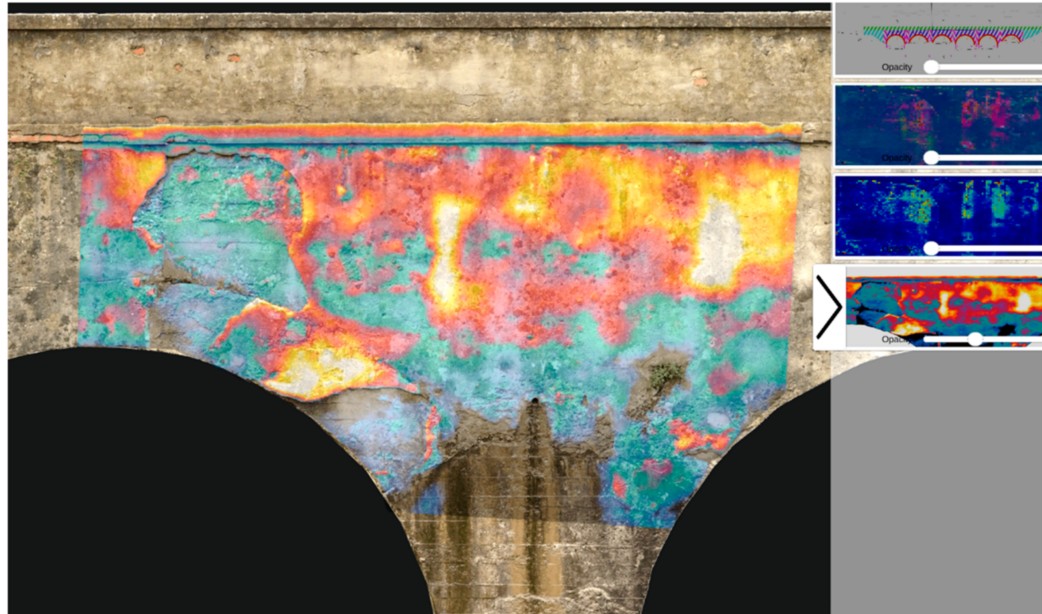

**Figure 3.** TOSCA-FI Software Platform: 2D heatmap overlays on a 3D digital bridge model [60].

### 3.5. Gravity Methods

Gravity Surveys (GS) use portable gravimeters [63], placed at regularly spaced sampling locations, to measure subtle variation in gravity surrounding railway tunnels [64]. Anomalies observed in returned Complete Bouguer Anomaly (CBA) curves inform subsurface material composition [65] and indirectly, structural health assessment. Regional trends in subsurface density conveyed by CBA curves can vary across scales comparable to the tunnel itself, granting extensive inspection coverage. Likewise, localised negative field displacements can indicate the presence of irregular low density regions, strong indicators of voids and deformation zones [66–68]. However, few common defects exhibit substantially large density variations (compared to their surrounding landmass) that would noticeably influence a CBA curve, which despite informing the general nature of the subsurface, does not comprehensively nor clearly visualise subsurface features themselves. Moreover, localising large features relative to the tunnel (i.e., in front, behind, left, right) is further complicated by the structure's cylindrical profile. This makes modelling the corresponding gravity field a multi-solution problem, introducing significant uncertainty and greatly increasing involved computation efforts [69].

### 3.6. Radar Methods

Ground Penetrating Radar (GPR) directs radio pulse emissions at the tunnel intrados, which penetrate and partially backscatter off strong dielectric gradients in the subsurface associated with features of interest [10,70–73]. Pulsed Radar (PR) samples consecutively emit wideband waveforms to measure backscatter in the time domain. Step-Frequency Continuous Wave (SFCW) radar incrementally sweeps an emission sinusoid through a predefined frequency band; the Fourier Spectrum of the returning signal is directly ascertained in the frequency domain by frequency-wise inspection of return signal strength [74]. In RTI, systems fall under three categories:

- **Trolley-Mounted** [75–77] (Figure 4a)—Units commonly feature interchangeable air-coupled antenna of differing frequencies to facilitate trade-off between penetration depth and output image resolution [28]. However, motorisation is infrequent, scans are unidirectional (typically railbed only) and offer no protection to operatives;
- **Handheld** [78–81]—Compact ground-coupled scanners guided by hand can achieve real-time scanning of curved tunnel sidewalls and crown. Typically restricted by limited penetrative depth (d < 50 cm), coverage speed (under m$^2$ h$^{-1}$) and gantry requirement to reach high surfaces make units impractical for full RTI;
- **Vehicle-Mounted** [82–86] (Figure 4b)—Multidirectional fixed antenna units attached to locomotives, rolling stock or RRVs. Although capable of data capture at speeds ranging from to 50–30 km h$^{-1}$, fixed directionality guarantees blind spot and air-coupling reduces achievable penetrative depth.

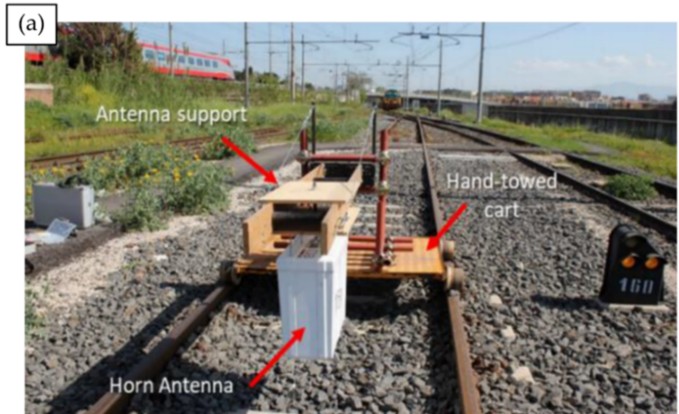

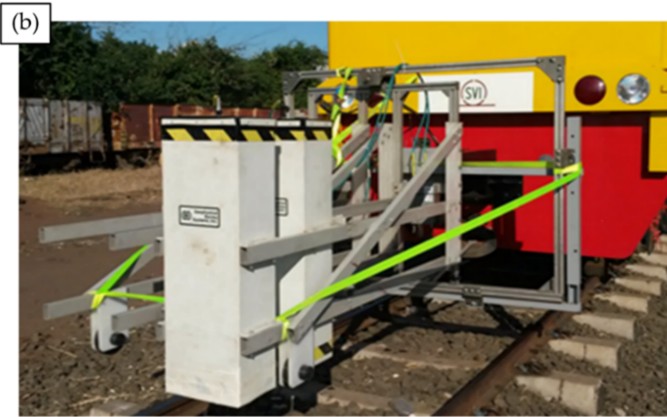

**Figure 4.** A selection of leading GPR systems. (**a**) Ballast fouling inspection trolley with selectable-frequency GPR antenna module [77]. (**b**) Loco-mounted fixed directional air-coupled GPR antennas.

Radargrams (B-scans) traditionally convey survey output, which although encoding multiple feature characteristics (e.g., depth, extent, orientation, heterogeneity, etc.) [70,73], demand extensive digital filtering [87–91] and may still contain abundant false-artifacts (e.g., ringing effects from rails, airwaves from overhead surfaces) [92–94], making them notoriously unintuitive. Increasing clarity via orthogonal intersection [95,96] (Figure 5a) and parallel stacking [97,98] (Figure 5b) of B-scans to form C-scans is now well-established practice in many commercially available GPR processing software packages [99–103]. Collectively, we denote this pseudo-3DGPR. We stipulate 'pseudo' to emphasise the inherent information loss resulting from 2D projections of 3D tomography. Beneficially, C-scan datasets can exploit time-slicing [104–106] (Figure 6a,b), in situ transparency filtering [107] and false-colouration [108] to improve conveyance of 3D forms. More recent investigations into true-3D volumetric reconstruction [98,109–117] (Figure 6c) show promise for advances

towards practically viable fully immersive GPR-based subsurface inspection surveys (undertaken in fully digitised virtual survey environments) [80,118–122]. Whilst conceivable and under trial, achieving mainstream commercial deployment will require blind spot alleviation through adoption of rotary scan motion complementary to tunnel curvature, possibly similar to the superposition of concentric cylindrical 'look-ahead' radargrams pioneered by the TULIPS system [123] for tunnel excavation monitoring. Pursuit of blind spot elevation is therefore of critical importance if comprehensive RTSSI profiles are to be captured.

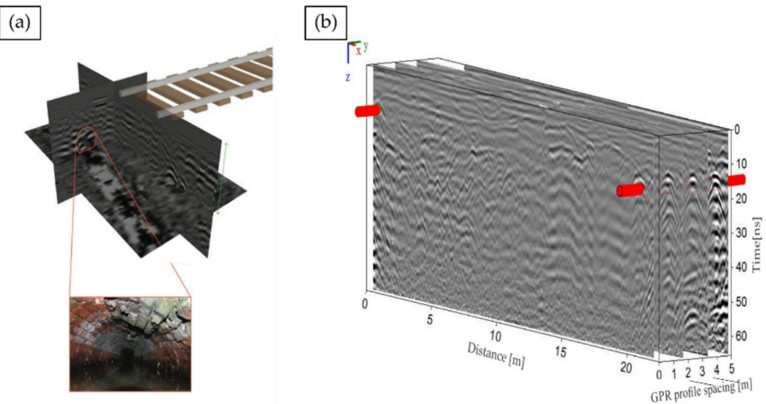

**Figure 5.** Principal methodology for combining planar radargrams to form pseudo-3DGPR visuals. (**a**) Orthogonal Intersection: B-scans meet at 90° helps focus attention on the central region. (**b**) Parallel Stacking: Aligning B-scans as slices of a cuboidal volume helps identify reoccurring targets [97].

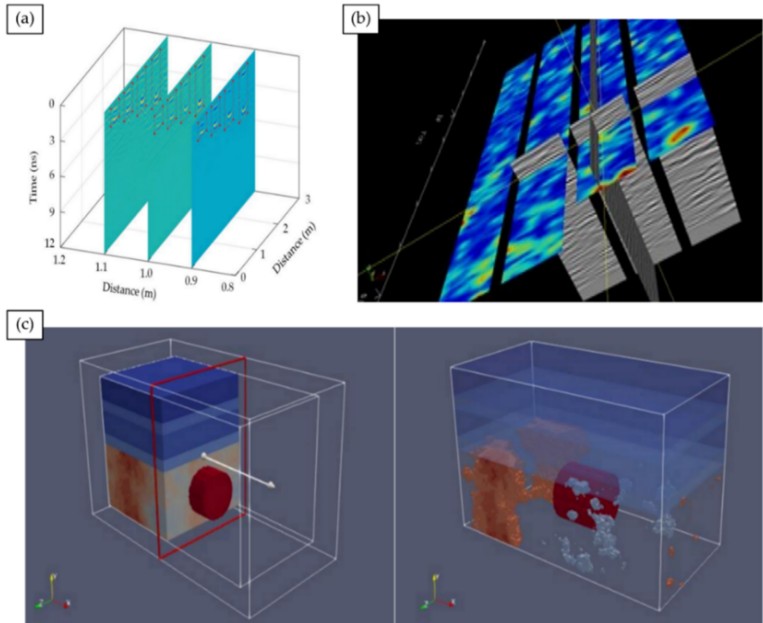

**Figure 6.** Mechanisms for improved contextualisation of 3D GPR datasets. Time-slicing performed (**a**) horizontally [100] or (**b**) vertically [102] can improve perception of relative depths and 3D feature shape. (**c**) Volumetric reconstruction increases feature faithfulness to reality, reducing human memory dependency, but processing remains highly involved [103].

### 3.7. Robotic Methods

Robotic systems reduce necessary human involvement in RTI, thereby beneficially reducing human error during data acquisition (e.g., mis-recordings due to subjectivity or

lapses in concentration). Scope for varying degrees of autonomy further reduces dependency on onsite human presence, thereby increasing crew safety and cutting overhead costs. However, we must remember robotic methods share the limitations of their constituent sensors and also exhibit their own unique set of challenges (e.g., collision avoidance, recovery, stabilisation, power management, miniaturisation).

### 3.7.1. Unmanned Aerial Vehicles

Unmanned Aerial Vehicles (UAVs) have become increasingly popular for tunnel inspection owing to their low cost designs, programmability and exceptional manoeuvrability, which has motivated in excess of $4 billion global investment in UAV technology development for infrastructure inspection [124]. However, practical performance of current UAVs remains limited by poor onboard charge retention [125]; stabilisation challenges from near-wall turbulence and common dependency on GPS. Note that being subterranean, Global Positioning Systems (GPS) typically struggle to operate reliably in tunnels. [126,127]. Although, we did find considerable recent research applying collision-aversion protocol [128,129] and 'smart pathfinding' [130–132] (e.g., PLUTO [133]) to develop autonomous UAVs [134–137]. However, backup pilots remain necessary which add costs and safety-risks [137]. Furthermore, no commercially available autonomous UAV has yet to be developed specifically for RTSSI, despite similar systems existing for hydroelectric penstock surface level inspection [138].

Being airborne, UAVs could quickly transport RTSSI sensors where articulated booms cannot reach, for instance UAV-SWIRL hovers inside vertical ventilation shafts [125,139]. However, most systems still favour Optical Photometry and LiDAR sensing [140–142], permitting only implicit subsurface measurements. Of novel importance, we discuss several significant exceptions developed since 2015. These include development of new UAV-mounted GPR prototypes [143–145]; we found one commercial system [146] capable of 10 m penetration, however it is unclear if this incorporates UAV altitude.

In addition, hybrid locomotion UAVs now encompass:

- **Fixed Anchor-Point Docking** [147–150]—Sustains surface contact for IST and UST but requires highly involved pre-installation of anchors;
- **Pivoting RTUs** [151] (Figure 7)—Tracks enable uninterrupted contact and continuous one-way surface coverage but increase weight and power drain, limiting survey completeness;
- **Negative Pressure Wall-Climbers** [152,153]—Faster than track-based RTUs but require large flat contact surfaces, hence curved tunnel geometry risks UAV slip and hazardous control loss;
- **Fully Actuated Configurations** [154,155]—Provides best all-round solution, providing unrestricted multidirectional movement even on curved surfaces, but is liable to near-surface turbulence.

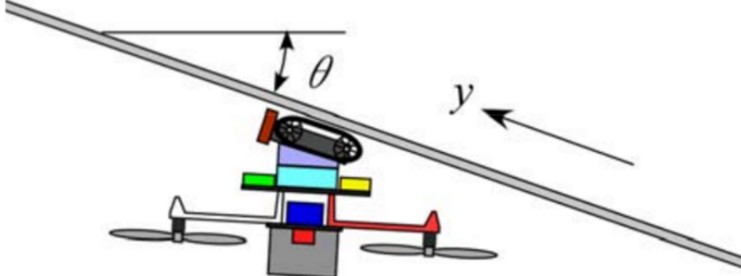

**Figure 7.** A recent novel innovation in hybrid locomotion UAVs. The pivoting traction crawler UAV performs IST on angled infrastructure surfaces inaccessible to engineers without gantries [151].

### 3.7.2. Adaptive Robots

We consider adaptive tunnel inspection robots to be devices capable of automatic geometry, operation or locomotion mechanism modification that combats demanding environmental conditions. As developments found were primarily proof-of-concept prototypes, current systems lack direct applicability to RTSSI without significant refinement efforts.

Nonetheless, we shall consider how such systems could be of practical future benefit in RTSSI. Foremost, adaption permits infiltration of inaccessible survey areas (e.g., drainage pipe interiors, capped shafts), increasing survey coverage. Moreover, units can swiftly traverse complex terrain (e.g., steps, rail tracks, damaged surfaces, angled walls) without human interaction, inviting remote inspection innovation potential.

Reconfigurable UAVs [156,157] fold (Figure 8a) to pass though narrow channels before unfolding (Figure 8b) to survey unknown void-like environments, which could be applied to preliminary surveys of hidden shafts via small diameter drill holes in capping facades. However, with more moving parts, damage likelihood during transit or execution is increased, potentially trapping systems behind walls incurring excess repair, replacement or recovery costs. Self-disassembly [158] could provide an easier route towards recovery.

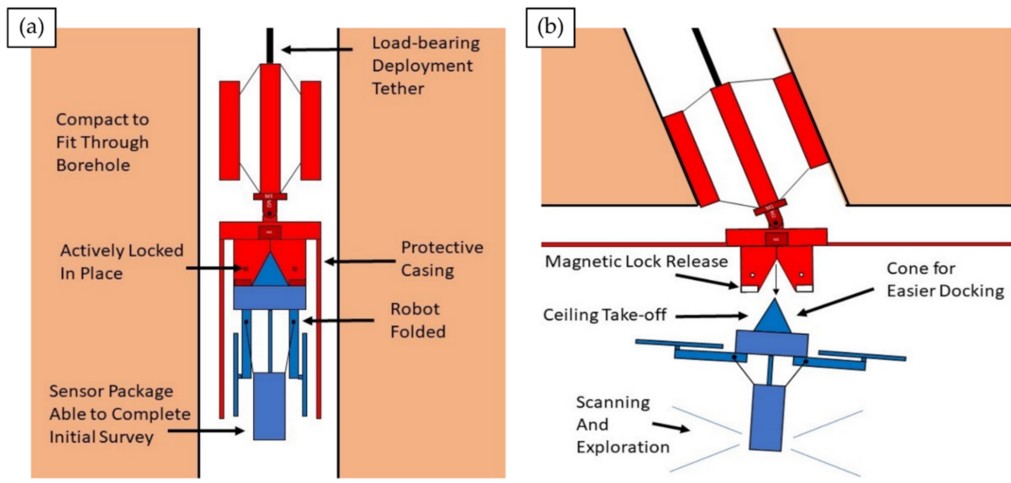

**Figure 8.** Folding UAV PROMETHEUS for subterranean inspection [156] can enter boreholes (**a**) to explore inaccessible voids (**b**) of potential benefit for probing hidden shafts in RTSSI.

Burrowing inspection devices [159] could create subsurface channels, then deploy 'snake robots' [160,161] fitted with endoscopes or fibrous sensing elements to directly image subsurface condition, detect ground movement [162] or moisture content [163]. However, burrowing is destructive and could exacerbate damage to already defective quadrants of intrados. Alternate use of modular configurations [164,165], compact step-climbers [111,166,167] or deformable 'soft robots' [3,168,169] fitted with NDI sensors could traverse small but pre-existing subsurface channels (e.g., pipes, vents, data cables) avoiding destructive burrowing. Soft robots uniquely could contort to bypass obstructions for multidirectional inspection of clogged drainage pipes. However, extensive development remains necessary to form a coherent self-arrangement of modular robots [170–174] capable of emulating established NDI techniques.

### 3.8. BIM-Integration

Building Information Modelling (BIM)represents a new paradigm for large structure lifecycle information management [175,176]. Current survey outputs represent one-way information exchanges between the physical tunnel environment and reconstructed digital models. By contrast, Digital Twin Tunnel (DTT) BIMs would facilitate two-way information exchange from any point in time during its perpetual update cycle. In two-way exchange, state changes in physical tunnel prompt reactive changes in the digital tunnel informing

future maintenance, which cause further state changes in the physical tunnel and so on and so forth [177].

We find multiple recent experimental rail tunnel-BIM studies exist [178–182], typically deploying laser methods to profile and categorise trackside assets (Figure 9) but only [183] directly approaches RTSSI, developing a prototype AI-assisted BIM for ingress detection (developed on the Amber Inspection Could). Problematically, none currently exhibit adequate automation to be considered idealised DTTs. By inference, visualisation quality and overheads would clearly benefit from the significantly increased data pools and optimised network architectures anticipated [184]. However, challenges remain. Existing BIM architectures frequently lack specialisation to account for unique RTSSI challenges, such as complex terrain deformations and changeable subsurface geological conditions [182,185]. Reoccurring incompleteness of feeder data from NDI methods further limit BIM efficacy for RTSSI, despite recent improvements in multi-label datasets recovery [186–189].

Evidently, BIM integration for RTSSI will be essential for developing the first self-sustaining DTT [188], but insufficient without complimentary improvements to survey completeness.

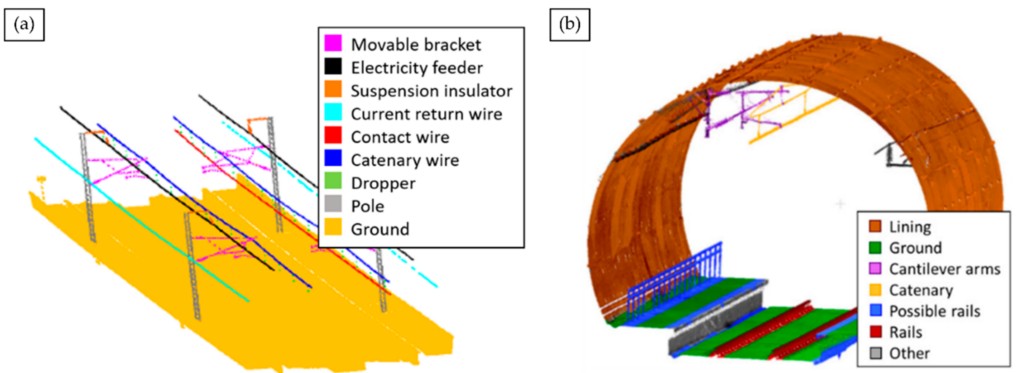

**Figure 9.** BIM for RTI. 'As-is' BIMs [181] frequently adopt LiDAR point cloud segmentation (via RANSAC and Supported Vector Machine methods) to label trackside assets both (**a**) outside and (**b**) within the tunnel environment [41].

### 3.9. Other Methods

Aforementioned NDI methods are most commonly deployed in routine RTSSI surveys based on encountered literature, motivating distinction from (i) more antiquated methods (e.g., invasive, inefficient, overly localised), (ii) less established experimental practices and (iii) schemes for real-time subsurface monitoring. In (i), we group: Borehole/Drill Core Sampling [190,191], Electrical Resistivity Tomography (ERT) [192–195], Endoscopic Probing [196,197] and Schmidt Hammer Strength Testing [53,198]. In group (ii), we gather: Radiography/Muon Tomography [199,200] and multiple additional prototype robotic RTSSI systems [14,166,201–204]. Group (iii) accounts for Time Domain Reflectometry schemes [205,206] and other Embedded Sensors [207].

### 4. Heuristic Comparisons of RTI Methods

Having discussed key attributes of leading RTI methods for RTSSI in isolation, we now direct the reader to our more comprehensive summary provided in Appendix A. It is tempting to directly compare advantages and disadvantages, 'ranking' methods to find an 'optimal' choice—a process we'll term 'heuristic comparison'. Noting that railway networks must balance inspection funding, duration and result quality, whilst researchers similarly prefer to invest effort in developments that return greatest impact, both for practical surveys and advances to the research field. This optimisation problem initially appears well-posed but this is not the case.

We find heuristic comparisons lack natural scaling. We may regard 'scaling' as a fixed-reference, quantifiable metric for comparing the importance of two characteristics. We draw parallels with use of numbered scales on questionnaires gauging attitudes (e.g., perceived risk between different dangers) [208]), therefore frequently suffer from:

1. Ambiguity maintaining a consistent comparative ground throughout;
2. Contextual variation between significance of comparative grounds.

Consistency ambiguity typically arises during first evolution of an argument (e.g., spoken discussion in planning meetings):

*"Let's compare the accuracy of subsurface 3D visuals produced by LiDAR and pseudo-3DGPR. The latter are clearly more accurate because LiDAR can only indirectly visualise the subsurface (cross-sectional deformation). But the former is more accurate because deformation appears as point cloud deformation, whereas physical features are not actually hyperbolae-shaped as they're shown in pseudo-3DGPR".*

Note that both comparisons are valid and concern accuracy, but lack a definitive conclusion. The comparison ground for 'accuracy' subtly shifts from data-type to data-faithfulness. We argue the origin is vague definition of what constitutes an 'accurate' visual in the comparison posed. By contrast, contextual variation is more obvious:

*"The spatial resolution of gravitational surveys would be inferior to Thermography for detecting small voids in an operating tunnel, but superior for strata mapping during construction".*

Evidently, a more robust rationale is required.

## 5. Criteria for an 'Effective' RTSSI Visualisation Framework

So far, we have found heuristic arguments unsatisfactory for appraisal of RTI methods in a RTSSI context. Significant disparity exists between respective operating principles, deployment methodologies and output conveyance; not least in the inherent multifaceted and context-dependant grounds for suitability and performance comparison. Two specific examples would include: (i) the nature/variety of detectable features and (ii) assessment timescale.

The basis for our criteria is twofold.

First, we recognise each RTI method discussed exhibits at least one critical limitation (Table 1). Logically, an effective RTSSI visualisation framework would not share any, meaning an innovation directly addressing either would have significant impact on the current RTI hardware market. This motivates our novel formulation of well-defined but sufficiently general criteria for an 'effective' RTSSI visualisation framework. We illustrate the benefits visually using a conceptual network diagram (Figure 10).

**Table 1.** Main current issues facing NDI methods.

| Method | Critical Limitations for RTSSI | |
|---|---|---|
| Visual<br>Laser | Lack necessary penetrative capability to directly visualise subsurface. | |
| Therm.<br>Acoustic<br><br>Radar | Information loss (2D projections of 3D features) limits survey faithfulness. | Skew from ambient temperature variations.<br>Inefficient implementation.<br>Curvature induces blind spots.<br>Visuals lack interpretive clarity. |
| Gravity | Struggles to resolve localised features. | |
| Robotic<br><br>BIM-Int. | Systems share the limits of their ancillary sensors. | Onsite supervision still required.<br>Majority of systems are still concepts or early prototypes.<br>Architectures frequently lack<br>sufficient optimisation to react to RTSSI data dynamically. |

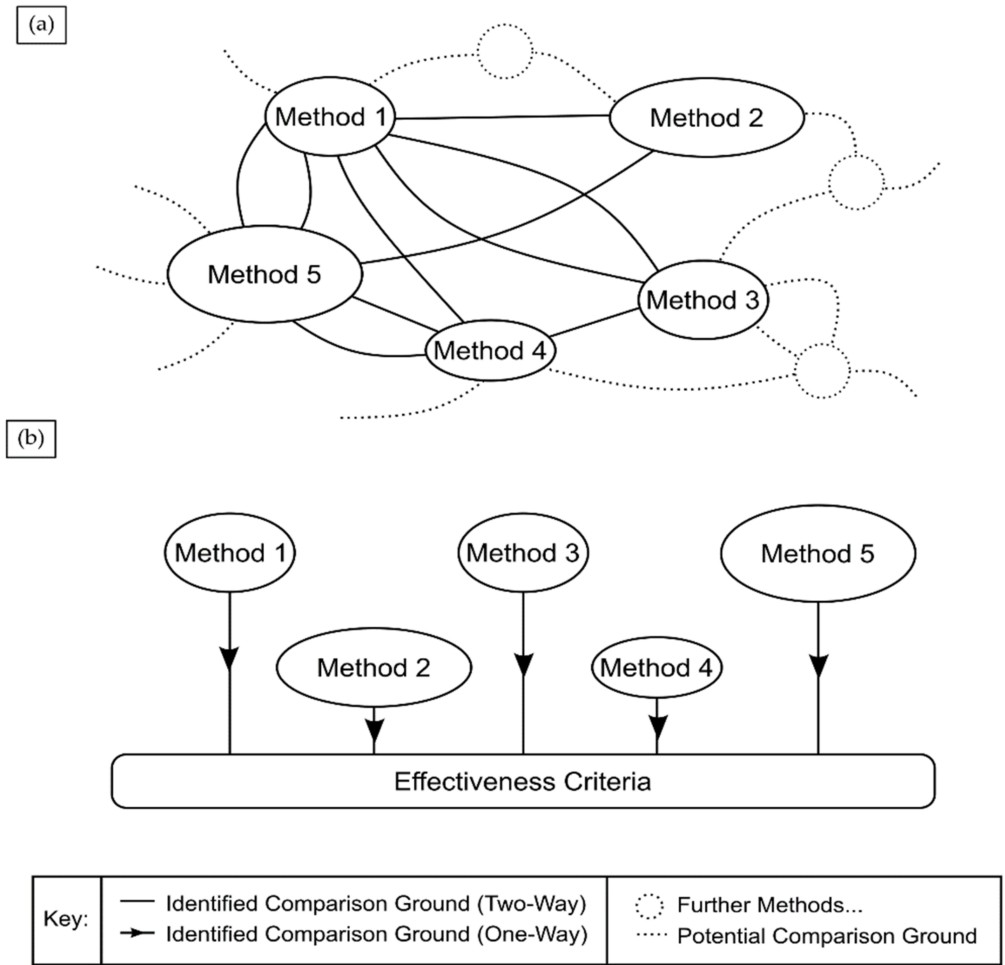

**Figure 10.** Using criteria greatly simplifies comparison networks. (**a**) Heuristic comparisons are irregular and multi-directional networks. (**b**) Criteria inclusion collapses the network to be regularised and unidirectional.

Second, our creation of a Category Connection Matrix (CCM) from encountered literature (Table 2)—inspired by the hybrid workflow matrix presented in [209])—highlights emerging research trends, which we use to infer future research avenues in RTSSI. The categories extracted inform where research attention is currently most directed. Furthermore, collectively, motivation for all relevant studies is to contribute to producing the most effective RTSSI visualisation framework possible. Thus, each identified research category must align with at least one criterion.

From this, we formulate our proposed criteria for an 'effective' RTSSI visualisation framework, which considers:

Data Acquisition:

1. **Completeness**—Uninterrupted scan coverage should be achieved to record the full extent of the influential regions of the tunnel subsurface. Current methods either lack deep penetrative capability or exhibit bind spots due high localisation of scans or geometry curvature.
2. **Duration**—Survey execution should balance acquisition speeds with recovered data quality (i.e., resolution, distortions) ensuring inspections and repairs cause minimal network disruption. A rapid low-quality scan limits inspection closure, but misinformed repairs take longer to fix and vice versa.

Information Conveyance:

1. **Accessibility/Interpretive Clarity**—A railway network end-user who is not a specialist in the utilised RTSSI technique(s) (e.g., planner) should independently be able to understand and make informed decisions based on visualisation output (consider radargrams the antithesis to this, containing considerable but mostly incomprehensible information).
2. **Faithfulness**—Inconsistency between the physical subsurface geometry undergoing inspection and corresponding representation within the visualisation medium should be kept to a minimum. An example of unfaithful conveyance is how overhead structures can confusingly appear as below-ground features (airwaves) in radargrams [93,210].
3. **Interactivity**—Visualisations should react intuitively to end-user engagement in ways that make surveys more ergonomic, efficient and versatile. Again consider radargrams; time-slicing in pseudo-3DGPR conveys depth more ergonomically than viewing am isolated B-scan.

Understandably, developing fully effective frameworks will take time, but we can make informed predictions. In consulting phase (1) literature; current GPR technology offers greatest versatility for RTSSI. Hence, we anticipate earliest industrial impact will likely stem from its unification with pre-existing conveyance innovations such as interpolative [111,116] and AI-assisted 3D feature recovery (e.g., DepthNet [114]).

**Table 2.** Category Connection Matrix (CCM).

| Categories | Identified Research Avenues | | | | | | | |
|---|---|---|---|---|---|---|---|---|
| Method | A | B | C | D | E | F | G | H |
| Visual | ● | ◐ | ◐ | ◐ | ◐ | ◐ | ◐ | ● |
| Acoustic | ○ | ○ | ○ | ● | ◐ | ○ | ○ | ○ |
| Laser | ◐ | ◐ | ◐ | ● | ○ | ◐ | ◐ | ◐ |
| Therm. | ○ | ○ | ◐ | ● | ◐ | ○ | ○ | ○ |
| Gravity | ○ | ○ | ◐ | ○ | ○ | ○ | ○ | ○ |
| Radar | ○ | ◐ | ◐ | ○ | ◐ | ◐ | ○ | ◐ |
| Robotic | ● | ◐ | ○ | ● | ○ | ○ | ○ | ○ |
| BIM-Int. | ○ | ○ | ○ | ○ | ◐ | ○ | ◐ | ◐ |

Avenue Codes: A: Autonomous Tunnel Surveys; B: Alternatives to Fixed-Direction Sensor Arrays; C: Surface-Subsurface Tunnel Survey Fusion; D: Automated Tunnel Feature Detection; E: Tunnel Subsurface Feature Severity Ranking; F: Volumetric Tunnel Feature Reconstruction; G: BIM/DDT Development; H: XR/RTI Integration. Icons: (○): Indicates a literature gap due to critical method limitations or a currently unexplored research avenue. (◐): Indicates works connected to an RTSSI research avenue exist but are indirectly related, signifying opportunity for new research via novel idea synthesis either amalgamated from or inspired by present literature. (●): Indicates works connected to a RTSSI research avenue exist and are directly related, signifying relevant practical research is proposed, presently underway or considered surplus to requirement.

## 6. Steps towards Criteria Fulfilment

Relevant research is already underway targeted at fulfilling our outlined criteria for an effective RTSSI visualisation framework.

Regarding the completeness criterion, we draw attention to the significant recent development of rotating, air-launched GPR antenna by Railview Ltd. (UK) [211] (Figure 11). Compared to fixed array GPR systems, including the Zetica Advanced Rail Radar (ZARR) [83,212] and the IDS SafeRailSystem (SRS) [84], the helical scanning trajectory of rotating antenna more closely mirrors naturally curved tunnel geometry, facilitating more comprehensive 360-degree RTSSI imaging, at competitive depths.

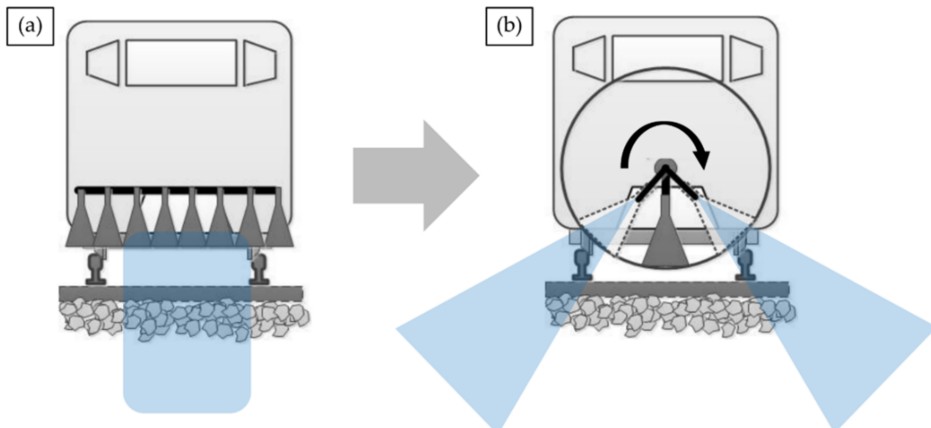

**Figure 11.** Unlike conventional fixed direction antenna (**a**) which typically only image the railbed; rotating GPR antenna capture 360-degree subsurface profiles including tunnel sidewalls, haunch and crown (**b**).

Lastly, we consider the conveyance criteria. Helical scans directly capture 3D RTSSI geometry in situ as 360GPR datasets, more akin to laser-methods than pseudo-3DGPR (requiring B-scan stacking or intersection). Ergo, forthcoming analysis of 360GPR presents an interesting opportunity for new research into volumetric feature reconstruction for RTSSI.

Thus, 360GPR has scope to form a visualisation framework meeting at least three of our five effectiveness criteria, feasibly disrupting the current RTSSI hardware market. At this point, owing to the larger proportion of relevant literature focused on addressing our conveyance criteria, we discuss the main innovations this review encountered into feature identification within subsurface datasets (Section 6.1) and dynamic interaction between visualisations and end-users (Section 6.2).

### 6.1. Automated Feature Detection and Evaluation

If an end-user cannot clearly interpret subsurface data, the inspection yields little useful insight into tunnel structural health or targeted maintenance. Detecting degradation indicators is critical for localising damage, whilst characteristic evaluation (e.g., location, extent, maximal depth, etc.) informs repair urgency. Searching RTSSI visuals manually is impractical: tunnel datasets are cumbersome; defect types are wide-ranging through a tunnel's operational lifespan; human cognition speeds are slow and our evaluation is subjective. Unintuitive data visualisations only compound the issue (e.g., radargrams), explaining why research tackling Automated Feature Detection/Evaluation (AFD/E) accounts for over 1/3 of CCM-featured research connections and encompasses Convolutional Neural Network (CNN) [213,214] and Deep Learning (DL) [49,215,216] detectors, alongside severity ranking schemes [217,218]. Scope of feature variety and complexity current AFD can simultaneously identify with accuracy drew our attention. Restricting our consideration to subsurface studies, a DL image grid workflow flags four distinct features [96] (manhole, cavity, pipe, heterogeneous soil background), yielding widest feature detection scope, albeit not concurrently. Studies successfully achieving simultaneous detection of realistically complex defect configurations [219,220] likewise favoured 2D GPR imagery but discriminated two types maximum [81]. However, with exception to [96], test environments featured only assets or defects. This implies GPR-based research is currently leading developments in AFD and seemingly the upper limits of DL feature detector capability have yet to be fully explored. Thus, any study classifying over four mixed type (assets, defects), variety (shaft, void, pipes) or complexity of features would mark a significant advance in RTSSI-AFD.

AFE, namely defect severity ranking, proves less researched. Contrary to our initial expectations of exclusively dictionary-based schemes, of studies found, most now adopt contextual evaluation via fuzzy logic devices [221,222] and probabilistic analysis [33,223]. We infer more complex evaluation grounds are being considered in parallel when grading repair urgency, if not yet for RTSSI. For example, many nearby cracks in close proximity can be of greater concern than one occurring in isolation. Collectively, this suggests development of a robust contextual severity ranking scheme for RTSSI would be of worthwhile pursuit in future research.

ADF/E is clearly transitioning from proof-of-concept simulations to practical deployment tests. In RTSSI, schemes will need to discriminate tunnel assets from more hazardous defects, therefore training demands programmer knowledge of common features. For already aged masonry tunnels, we found no consolidated summary, concerning given a forecast 30–50% increase in rail-traffic demand by 2050 [224]. We therefore now present our own bespoke consolidated summary.

6.1.1. Common Assets in Masonry Railway Tunnels

An 'asset' denotes any useful or valuable item associated with railway network operation, encompassing employees, track, signalling, buildings, utilities and civils (structures, earthworks) [225]. Tunnel assets fall under civils, with any unprotected structurally significant entity designated a critical element.

Hidden Critical Element (HCE) is unobservable from at least one side [226]. Locating HCEs has presented considerable challenges for Network Rail (UK) in RTSSI. We identify that for current inspection methods, greatest challenge is presented by detection of blind (concealed but disenable) and hidden (concealed and indiscernible) shafts:

a.  **Ventilation Shafts**—Hollow columns extending from tunnel crown to the surface. They facilitate air circulation and were originally used to remove material during construction [227].

b.  **Maintenance Shafts**—To allow simultaneous excavation of multiple faces, many shafts would be sunk along proposed tunnel routes [228]. Typically infilled or converted to ventilation shafts, capping frequently conceals them for aesthetics (processes rarely recorded in writing). Being unreinforced, many have deformed or partially collapsed.

This is most true for Wales and Western Regions of the UK, following Network Rail's failure 'to deliver on a commitment to identify all hidden tunnel shafts by the end of 2016–2017' [229] and more recent delays tackling HCE examination schedules in 2019–2020 due to pandemic impacts [230].

As cavities are prime sources of water infiltration, failure to identify hidden shafts significantly increases risk of accelerated compromise to surrounding structure. Therefore, innovations towards hidden shaft detection present opportunity for highest new research impact in RTSSI.

Training detectors to discriminate hidden shafts from other features motivates continued summary of other common masonry railway tunnel assets:

c   **Overhead Line Equipment (OLE)**—Furthermore, dubbed 'traction wires' or the 'catenary', these high voltage electrical pickup lines power electric locomotives via onboard pantograph connectors. Systems can be integrated into older tunnels during electrification works. Forms include tensioned metallic cables mounted to the crown and the Rigid Overhead Conductor Rail System (ROCS) [231], which provide more efficient operation in low-clearance tunnels. Structural weakening can result from necessary drilling during install, whilst strong electromagnetic fields generated by the power feed can interfere with data acquisition systems and present line of sight obstruction during haunch or crown inspections.

d   **Portals**—Reinforced surfaces surrounding tunnel entrances combating outward deformation induced by shear stresses from continuous shifting of landmass encircling the tunnel [232]. Exposed to the elements, portal rigidity deteriorates, risking collapse

if cracks and displacement are not detected early. Reinforcement schemes include buttresses, ground anchors, and steel mesh coverage fixed with soil nails [233].

e　**Refuges**—Small arched recesses within the tunnel lining to protect railway workers from locomotives.

f　**Buried Utilities**—These can include both metallic and plastic water drainage pipes [234], electrical wiring and telecom cables.

g　**Trackside Objects**—These include signage, signals, electrical junction boxes and CCTV units.

h　**Culverts**—Small passages allowing watercourses to pass under railway tracks, including underground rivers [235]. Old masonry culverts particularly can be weakened by solution and hydraulic action resulting in partial section collapse, deforming the railbed above and causing water backlog which floods tunnels.

6.1.2. Common Defects in Masonry Railway Tunnels

Defects constitute any imperfections in the material or form of a structure. Indicative of degradation and increased failure likelihood under strain. Swift detection in an enclosed tunnel environment is critical. We provide a brief overview of causes and hallmarks for common defects in masonry railway tunnels, then reflect on the efficacy of their detection in modern surveys, highlighting any opportunities for future research:

a.　**Arch Barrel and Cross Sectional Deformations** (Figure 12a)—Shifting tunnel landmass induces changeable tensile and compressive forces within arch barrels. This can trigger sidewall bulging and buckling; haunch distortion; tunnel floor bowing or side-offset of the crown.

b.　**Cracks and Fracturing** (Figure 12b)—Localised shear forces and vibrations from rolling stock can split and displace masonry. Damage ranges from hairline cracks lacking obvious signs of displacement, to large open fractures exhibiting significant displacement.

c.　**Water Ingress** (Figure 12c)—Rain infiltrates tunnels through shafts and groundwater propagates through dissolved subsurface joints and fissures (karsts) [236]. Leaching of mortar begins as water percolates between masonry before flowing down the sidewall. Lubrication of masonry joints leads to movement of the structure, resulting in lining deformation. Owing to joint length, significant quantities of ingress can accumulate, scouring supports and flooding tunnels without proper drainage or saturated catchpits, as ingress often carries dissolved ochre (acquired during subsurface percolation) which forms crystalised limonite deposits [197] that block drains. Out-flow down sidewalls forms noticeable white streaks emanating from the region of breech, therefore can be used as indicators of ingress source. However, establishing subsurface mortar leaching extent is reliant on NDI methods for RTSSI.

d.　**Open Joints and Perished Mortar** (Figure 12d)—Characterised by the deterioration and eventual absence of mortar between brickwork. As brick tunnels can date back over 150 years, mortar naturally begins to deteriorate from reactions with moisture in the bricks, air and subsurface [227,231,237]. This process is accelerated by wash from nearby ingress, vibrations induced by both rolling stock and air-pressure waves from passing locomotives.

e.　**Loose and Missing Brickwork** (Figure 12e)—Early onset of spalling and failed patching repairs can result in loosened or missing brickwork, indicated by brickwork rubble on the railbed. Individually, gaps pose no substantial loss of structural rigidity, but can provide opportunity for larger defect growth if not addressed quickly. Furthermore, if present in tunnel haunch or crown, falling rubble can damage rolling stock, or cause serious injury to ground crews working below.

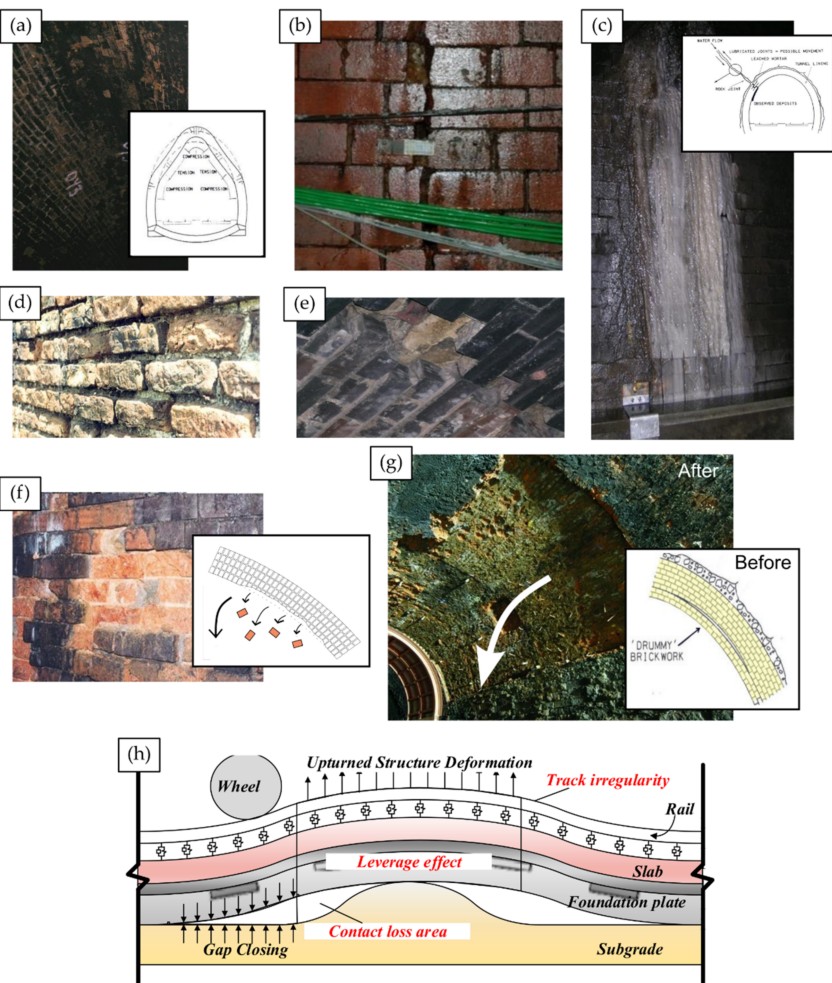

**Figure 12.** Common defects in masonry railway tunnels: (**a**) Haunch Deformation; (**b**) Cracks; (**c**) Water Ingress; (**d**) Perished Mortar; (**e**) Missing/Loose Brickwork; (**f**) Spalling; (**g**) Ring Separation and Debonded Wall-Section; (**h**) Frost Heave [238].

f   **Spalling** (Figure 12f)—Perishing masonry on the tunnel intrados in vicinity of the haunch or crown can be dislodged by gravity, leaving the next layer of brickwork exposed. Repetition gradually creates rough-profiled recessed quadrants. Bricks in newly exposed layers lack tarnishing from exposure to soot and locomotive exhaust fumes, thus a second hallmark is more vibrant brickwork colouration.

g   **Ring Separation and Debonding** (Figure 12g)—Arch barrels contain multiple layers of concentric brickwork rings. Literature encountered discussed brick tunnels ranging from 3–15 layers [237,239,240], evidencing significant possible contextual variation. Ingress, deterioration of intrados mortar and poor quality workmanship can all cause neighbouring rings to separate within the wall. Gravity pulls innermost layers downward, causing debonding from layers behind forming slit voids (subsurface hairline fractures). Over time, large voids begin to grow. If near-surface, separating rings may briefly cause visible cracking of intrados mortar, allowing detection. However, separations deeper than a single ring are completely invisible to an in-tunnel observer. With time, large sections of rings can debond, causing arcs of brickwork to fall away as slabs. Such 'delamination' events can deform rails or damage the railbed presenting a derailment risk. Depending on size, debonding can significantly weaken substantial volumes of surrounding masonry.

h   **Railbed Faults** (Figure 12i)—Subgrade layers of ballast facilitate even distribution of rail-traffic weight, ensuring rails remains level on uneven ground to prevents derailments. Displacement induced by ballast fouling [75–77] and frost heave [238,241,242]

can damage rails and offsets train weight distribution, increasing in-tunnel derailment risk and subsequent likelihood of major network disruptions.

i     **Drainage Faults**—Tunnels contain integrated pipework and catchment pits to safely remove excess water and silt. Flooding may occur if pipes become blocked, rupture due to freezing and expansion, or become overwhelmed by intense weather events [243]. Improvement works can also fail. For example, 18 bolts supporting a water catchment tray in Balcombe Tunnel (2011) decoupled due to resin failure [244]. Sag reduced tunnel clearance from 0.87 m to 0.3 m posing a dangerous obstruction to rail-traffic.

Upon reflection, we first note recent studies focus primarily on (and achieve) detection of three main defects: (i) surface-visible cracks via CRP; (ii) voids via Acoustics, Thermography or GPR; (iii) water ingress via Thermography and GPR. By extension, with suitable modification to enable multi-directional scanning, similar hardware could reasonably detect (i) open joints and perished mortar, spalling, missing brickwork; (ii) ring separation and debonding and (iii) drainage faults in future applications.

Secondly, we observe that currently cross-sectional deformation can only be directly imaged by laser methods as they boast 3D point cloud data capture. We note that traditional visual inspection also detects deformation, but notetaking is inherently far less accurate than imaging. Although no encountered literature directly utilised laser methods to detect railbed faults, we may reasonably assume cross-sectional imaging would also provide usable insight with relative ease.

Finally, we report that no data acquisition method is single-handedly able to detect each identified common RTSSI defect. This agrees with our findings in Section 3. Discounting Thermography since it cannot directly measure target depth, we remark that GPR has capability to detect the widest range of defect types (five of nine). Note that direct arrival waveforms are liable to mask small surface level defects, hence are not included in this statistic.

To improve this ratio, we believe future research into multi-sensory data acquisition systems presents high potential impact. A unit amalgamating RGB-CRP (surface imaging), LiDAR (cross-sectional profiling) and 360GPR (subsurface imaging) could feasibly detect all defects listed.

### 6.2. Extended Reality for Dynamic Survey Interaction

We define Dynamic Survey Interaction (DSI) to encompass any information conveyance technique that intuitively responds to end-user triggers (e.g., gestures, camera proximity, movement speed, metadata, field-of-view) [245–247], thereby increasing clarity of information within a survey and ergonomics of use. DSI attributes fundamentally reduce application complexity, boosting data usage efficiency and its accessibility to non-specialists, which had made them common features in Extended Reality (XR) interfaces.

Here, for clarity, we restrict our consideration of XR to just two subsets: Augmented/Mixed Reality (AR/MR) and Virtual Reality (VR). Note there is a distinction between AR and MR, AR is effectively passive information overlay, whereas MR is active, allowing the physical-environment to influence the digital environment and vice versa. In both cases survey data is conveyed through a head-mounted-display, which in AR/MR superimposes assistive digital information over the users' view-field, whereas in VR it provides full immersion in a digitally rendered environment [247,248]. As an emerging exploratory medium, academics and industry are now exploring practical applications of XR for DSI, including use for infrastructure inspections.

We encountered multiple noteworthy AR/MR visualisation developments outside phase (1). In [117], a rendering pipeline utilising back-projection of air-coupled 3D multistatic GPR data and Jerman Enhancement Filtering is presented but lacks an interface. Contrastingly, [249] demonstrates a prototype Unity3D-built AR pavement-subsurface visualiser for IOS based on 'Reality-Capture' modelling. However, performance of all subsurface AR inspection tools found appears unstudied in real railway tunnels. We anticipate

large volumes of subsurface data necessary for practical surveys will forgo local storage on commonly utilised tablets, requiring wireless relay from remote data-hubs. Herein we speculate the frequent absence of underground wireless communication networks in older railway tunnels may be responsible for research stagnation.

By comparison, several synchronisation schemes have been developed for real-time VR-BIM data exchange (e.g., BVRS [250]) and trailed in real tunnel construction projects (e.g., The Shenzhen-Zhongshan Immersed Tunnel [251]). For operating tunnels, literature concerning disaster situation training [252,253] dominates, whilst we found inspection studies to be scarce.

All but one VR visualisation framework was encountered (across [218,254]) tailored for RTI. The Enhanced Photorealistic Immersive (EPI) Survey Platform is developed in UE4 from SfM. Processed CRP data feeds a novel interactive dashboard to provide an extensive range of DSI attributes.

Techniques include: (i) defect highlighting filter toggles; (ii) a mini-map of in-model user location and (iii) proximity-triggered defect information modules detailing TCMI grading. The TCMI (Tunnel Condition Marking Index) ranges from 0 to 100, where 100 denotes a defect free aspect of the tunnel. Sadly dependency on visual CPR data does not facilitate subsurface inspection, undermining direct application to RTSSI.

Applying recent innovations in XR for DSI to RTSSI presents a promising direction for future research, which could significantly increase the clarity and accessibility to non-RTSSI specialist roles in tunnel management (e.g., asset engineers, environmental managers, operations risk advisors, etc.) [255–257].

The next key milestone facing AR/MR-RTSSI system deployment will be development of a dedicated wireless subterranean communication network supporting real-time information exchange with rail network data-hubs. We believe use of IoT/WSN Wireless Mesh mine communication nodes [257–259] could present a feasible solution and interesting research opportunity. For VR systems, we believe the lack of comprehensive subsurface datasets discussed in Section 5 explains the absence of research into a dedicated RTSSI application, whilst the visualisation framework presented in [218] showcases current state of the art in DSI for tunnels. We theorise future successes in 360GPR visualisation would open a practical research avenue into VR-based RTSSI.

## 7. Discussion

Method-centred subdivision of state of the art literature, spanning both data acquisition approaches and conveyance schemes (circa 2015–2021) reveals considerable recent advances in the capabilities of RTSSI-linked visualisation frameworks. Our review addresses two key knowledge gaps and presents three promising considerations for future research.

Appendix A summarises our deconstruction of leading NDI techniques for RTSSI, establishing that a multitude of valid comparative grounds exist between methods. Their variable importance subject to survey context undermines heuristic direct comparisons of method performance, suitability or efficacy by balancing advantages against disadvantages, resulting in ambiguity and inconclusiveness. This justifies need for a robust definition of an 'effective' RTSSI visualisation framework to address the knowledge gap. Our formulation of explicit criteria from five key research gaps was based on common shortcomings identified between considered methods. Respective criterions consider the (i) completeness and (ii) duration of survey data acquisition; alongside the (iii) interpretive clarity, (iv) faithfulness and (v) interactivity of information conveyance.

We overall find that despite recent innovations, a fully effective RTSSI visualisation framework has yet to be developed.

Creation of our CCM facilitated inference of eight key avenues for future research. Grading connected developments by relevance drew distinction between emerging and established trends within the literature. Initial thoughts towards achieving complete, time-efficient RTSSI surveys highlights pioneering analysis of 360GPR datasets from a novel ro-

tary, air-launched GPR antenna. Already meeting three criteria, we anticipate development of a suitable information conveyance scheme will present a prime research opportunity, with feasible scope for creating the first fully effective RTSSI visualisation framework within the next decade. This would establish 360GPR as a mainstream and potentially preferential technique amongst current RTSSI methods.

We note over 1/3 of research categories concerning information conveyance are aligned to automated target detection and defect severity ranking. However, their practical application on 3D datasets containing realistic quantities, varieties and complexities of subsurface features remains largely unexplored. We believe successful trails on authentic RTSSI datasets present an upcoming milestone for future research efforts to achieve if devised methods are to eventually aid real surveys.

Thoughts on detection scheme training for recognition of subsurface features in masonry tunnels flagged a further knowledge gap. Namely, this review was unable to find a combined nor comprehensive list of common masonry-related assets and defects within recent literature. Focus primarily centred on modern concrete tunnels. To address this, we presented our own bespoke consolidated summary for masonry tunnels.

Our exploration of emerging trends in dynamic interaction with current visualisation frameworks found studies addressing dataset interfacing with XR hardware featured 63% more prevalently than BIM/DTT system design. Most notable interaction potential was demonstrated by a VR tunnel surface survey platform presented in [218], which justified by end-user trails, quantifiably evidences the high levels of intuitiveness both 3D rendered environments and contextual dashboard modules can achieve. As with most encountered schemes, optimisation is for surface surveys only and no preview facility is provided during data acquisition. Arguably, this could also be beneficial; encouraging off site analysis of survey data in safer environments reduces crew risk.

Nonetheless, based on this review we believe VR presents the most versatile and intuitive tunnel survey interaction medium presently available, therefore would provide an ideal basis for future RTSSI visualisation framework developments. Application to a hybridisation of CRP, LiDAR and 360GPR datasets poses an interesting research opportunity and potential industrial solution for simultaneous surface and subsurface RTSSI surveys.

**Supplementary Materials:** The following supporting information can be downloaded at: https://www.mdpi.com/article/10.3390/app122211310/s1, for the literature summary tables referenced in Sections 3.1–3.8 and Appendix A, we direct the reader to consult the supplementary file <Literature_Summary_Tables.xlsx> uploaded alongside this manuscript.

**Author Contributions:** Conceptualization, T.M., M.R. and G.Y.T.; methodology, T.M.; software, N/A; validation, T.M., M.R. and G.Y.T.; formal analysis, T.M.; investigation, T.M.; resources, T.M.; data curation, T.M.; writing—original draft preparation, T.M.; writing—review and editing, T.M., M.R. and G.Y.T.; supervision, M.R. and G.Y.T.; project administration, T.M.; funding acquisition, T.M. All authors have read and agreed to the published version of the manuscript.

**Funding:** This work was supported by the UK Engineering and Physical Sciences Research Council (EPSRC) Grant EPT5179141 for Newcastle University.

**Institutional Review Board Statement:** Not applicable.

**Informed Consent Statement:** Not applicable.

**Data Availability Statement:** Not applicable.

**Acknowledgments:** The authors would like to thank Railview Ltd. (UK) for technical consultation on RTSSI hardware specification and provision of photographs presented in Figure 12.

**Conflicts of Interest:** The authors declare no conflict of interest. The funders had no role in the design of the study; in the collection, analyses, or interpretation of data; in the writing of the manuscript; or in the decision to publish the results.

**Abbreviations**

(ACT) Active Thermography; (AD) Adaptive; (AFD/E) Automated Feature Detection/Evaluation; (AR/MR) Augmented Reality/Mixed Reality; (BIM) Building Information Modelling; (CBA) Complete Bouguer Anomaly; (CCM) Category Connection Matrix; (CLSS) Circular Laser Scanning System; (CNN) Convolutional Neural Network; (CRP) Close-Range Photogrammetry; (DIFCAM) Digital Imaging for Condition Asset Monitoring System; (DL) Deep Learning; (DSI) Dynamic Survey Interaction; (DTT) Digital Twin Tunnel; (EPI) Enhanced Photorealistic Immersive; (ERT) Electrical Resistivity Tomography; (FA) Fully Autonomous; (GPR) Ground Penetrating Radar; (GPS) Global Positioning System; (GS) Gravity Surveys; (HCE) Hidden Critical Element; (IRT) Infrared Thermography; (IST) Infrasonic Testing; (LiDAR) Light Detection And Ranging; (MTPM) Moving Tunnel Profile Measurement; (NDI/E) Non-Destructive Inspection/Evaluation; (OLE) Overhead Line Equipment; (PR) Pulsed Radar; (ROCS) Rigid Overhead Conductor Rail System; (RRV) Road-Rail Vehicle; (RS) ROBO-SPECT; (RT) Rail-Trolley; (RTI) Railway Tunnel Inspection; (RTSSI) Railway Tunnel Subsurface Inspection; (RTU) Robotic Traction Unit; (SA) Semi-Autonomous; (SFCW) Step-Frequency Continuous Wave; (SfM) Structure From Motion; (SRS) SafeRailSystem; (TLS) Terrestrial Laser Scanning; (UAV) Unmanned Aerial Vehicle; (UST) Ultrasonic Testing; (VR) Virtual Reality; (XR) Extended Reality; (ZARR) Zetica Advanced Rail Radar.

**Appendix A**

For NDI methods discussed in Sections 3.1–3.8, we provide comprehensive summary tables of key literature analysed in this review, both for completeness and to evidence the multitude of valid comparative grounds on which heuristic comparisons may be based. The Literature Summary Tables are included as Supplementary Material.

Abbreviations adopted in the tables carry over from each section of the review. Below we define an additional visual scale to rank the interpretive clarity of systems presented, alongside a global legend of additional shorthand notation used exclusively in the Literature Summary Tables.

**Table A1.** Literature Summary Tables: Interpretive clarity scale.

| Symbol | Clarity | Necessary Training |
|--------|---------|-------------------|
| ○○○ | High | Training Not Required. |
| ○○● | Mid-High | Some Require Light Training. |
| ○●● | Low-Mid | Most Require Moderate Training. |
| ●●● | Low | Extensive Training Essential. |

**Table A2.** Literature Summary Tables: Shorthand notation.

| Column Field | Type | Shorthand |
|--------------|------|-----------|
| Status | Concept | C |
| | Prototype | P |
| | Commercial System | CS |
| Motion | Static | ST |
| | Handheld | HH |
| | On-Rail | OR |
| | Airborne | AB |
| | Crawler-Unit | CU |
| | Robotic Arm | RA |
| | Adaptive Traction Unit | ATU |
| | Pneumatic Suction Feet | PSF |
| | Tunnel Boring Machine | TBM |

**Table A2.** *Cont.*

| Column Field | Type | Shorthand |
|---|---|---|
| Duration | Seconds | S |
| | Minutes | M |
| | Hours | H |
| | Days | D |
| | Weeks | W |
| Key Target Types | Cross Sectional Deformation | CSD |
| | Hot/Cold Spots | H/C |
| | Groundwater Flow | GF |
| | Buried Utilities | BU |
| | Ballast Fouling | BF |
| | Trackside Assets | TA |
| | Power Distribution | PD |
| | Voids/Debonding | V/D |
| | Ore-Deposits | OD |
| Additional Symbols | Not Applicable | N/A |
| | Information Unavailable | - |
| | Important Note | * |

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
