# Peer review of "Developments in 3D Visualisation of the Rail Tunnel Subsurface for Inspection and Monitoring"

_applsci, doi:10.3390/app122211310_

Round 1
Reviewer 1 Report
I suggest that he pass the review and correction of the English languageAuthor Response
Please see the attachment.

Reviewer 2 Report
1. Grammatical and linguistic issues, like the one in line 19.
2. Revised title as the existing one is the reflecting the review.
3. Why specifically for railway?
4. Define the abbreviations for the first time in the text.
5. 300 resources? If the authors mean references, then they must check the actual number of references.
6. A flowchart for clear understanding the study methodology, application, scope, and limitation.
7. Summarize the application and limitation of existing leading subsurface data acquisition methods in tabulated form.
8. Conclusion?
Reviewer 3 Report
This is an excellent and timely review of the rapidly expanding field of 3D visualisation of civil infrastructure for inspection and monitoring. At the moment we are at an inflection point where the AI based automated and continuous (integrated sensor systems) approaches enter the mainstream. In fact, I was somewhat surprised as were the authors that defect severity evaluation has not got more attention in related research.
It is my pleasure to recommend this review paper for publication in Applied Sciences.
Minor comments:
1. Quotations should have citations.
2. There are some odd layout issues:
pg. 19 l. 600
pg. 26 l. 896
3. pg. 26 l. 872: 8 key avenues -> eight key avenues
Round 2
Reviewer 2 Report
.